# Automatic Generation of SBML Kinetic Models from Natural Language Texts Using GPT

**DOI:** 10.3390/ijms24087296

**Published:** 2023-04-14

**Authors:** Kazuhiro Maeda, Hiroyuki Kurata

**Affiliations:** Department of Bioscience and Bioinformatics, Kyushu Institute of Technology, 680-4 Kawazu, Iizuka 820-8502, Fukuoka, Japan

**Keywords:** GPT, large language model, kinetic modeling, simulation, systems biology

## Abstract

Kinetic modeling is an essential tool in systems biology research, enabling the quantitative analysis of biological systems and predicting their behavior. However, the development of kinetic models is a complex and time-consuming process. In this article, we propose a novel approach called KinModGPT, which generates kinetic models directly from natural language text. KinModGPT employs GPT as a natural language interpreter and Tellurium as an SBML generator. We demonstrate the effectiveness of KinModGPT in creating SBML kinetic models from complex natural language descriptions of biochemical reactions. KinModGPT successfully generates valid SBML models from a range of natural language model descriptions of metabolic pathways, protein–protein interaction networks, and heat shock response. This article demonstrates the potential of KinModGPT in kinetic modeling automation.

## 1. Introduction

Kinetic modeling plays a crucial role in systems biology, enabling researchers to quantitatively analyze and predict the behavior of complex biochemical systems, such as metabolic pathways and gene regulatory networks [1]. Kinetic models are differential equation models that describe the dynamic behavior of biochemical systems based on the interactions between their molecular components. Kinetic models are stored and distributed as SBML (Systems Biology Markup Language) files [2,3]. In addition, there are several software tools, such as Tellurium [4,5], COPASI [6,7,8], and CADLIVE [9,10,11], that support kinetic modeling. Still, the development of kinetic models is time-consuming and expert work. First, modelers survey research articles for known enzyme reactions, signal transduction pathways, and gene regulations in the system of interest, and build the chemical reaction equations with kinetic parameters. Next, modelers translate them into kinetic rate equations. A few studies have been devoted to developing natural language processing-based methods, that generate kinetic models from literature texts [12,13]. However, these studies focused on signal transduction, a specific type of biochemical system. Moreover, their complex implementation makes it challenging for non-experts to modify or extend.

In recent years, there has been significant progress in artificial intelligence (AI) technologies, particularly in the field of large language models (LLMs). GPT (generative pre-trained transformer) models stand out as the most advanced among these. ChatGPT [14], one of the GPT models, has gained public attention due to the ability to engage in natural conversations. It has been shown that GPT models have the capability to pass graduate-level exams [15,16,17] and write program codes [18]. In addition, GPT models may allow researchers to automatically build kinetic models directly from research articles. To the best of our knowledge, the capabilities of GPT models for automatic kinetic model construction have not been investigated.

This article aims to answer the question: Can a GPT model generate an SBML kinetic model from a natural language description? As GPT alone cannot create valid SBML models, we propose a novel approach called KinModGPT, which combines GPT as a natural language interpreter and Tellurium as an SBML generator. Moreover, we demonstrate that KinModGPT can generate valid SBML models from natural language descriptions of biochemical reactions.

## 2. Results

### 2.1. GPT Alone Cannot Create Valid SBML Models

First, we tested whether GPT models can directly generate an SBML model from natural language text. GPT models are LLMs trained on massive amounts of text data [19,20]. When prompted with a message to start the conversation, GPT models generate appropriate responses by repeatedly predicting the following word in the sequence. In this study, we tested three GPT models: *text-davinci-003*, *gpt-3.5-turbo*, and *gpt-4* (see Section 4).

We employed four test problems, with biochemical systems with different complexities (Figure 1). The first problem, decay, is the simplest scenario, in which an SBML model is generated from just two sentences. This model has a single variable and a single reaction. The HIV model represents the mechanism of irreversible inhibition of HIV proteinase [21,22,23]. It comprises nine variables and ten reactions, and its model description consists of five sentences. The three-step model describes a hypothetical metabolic pathway with three enzyme reactions and three gene regulations. This model consists of 10 variables and 15 reactions, described in 6 sentences. Finally, the heat shock response model represents the realistic, complex regulatory mechanisms that confer robustness to heat shock in *Escherichia coli* [24,25,26]. The heat shock response aims to refold proteins denatured by heat. This system revolves around the transcription factor $\sigma^{32}$, the chaperone protein DnaK, and the protease FtsH. The heat shock response model comprises 25 variables and 50 reactions, described in 20 sentences.

In the computational experiment, we asked the GPT models to convert each model description to an SBML model. The instruction message for the GPT models is shown in Figure A1. The results are summarized in Table 1. For all the model descriptions tested, the GPT models generated SBML-like models. However, upon inspecting these SBML models using the Online SBML Validator [27], we discovered that all the generated SBML models were invalid, containing some errors, such as missing required attributes. The invalid SBML files cannot be imported by widely used modeling tools. In summary, GPT models alone cannot generate correct SBML models for the four test problems presented in Figure 1, highlighting the difficulty of generating SBML models from natural language texts.

### 2.2. KinModGPT Can Create Valid SBML Models

To generate an SBML model from natural language text, we introduce the strategy named KinModGPT, as outlined in Figure 2. First, using a GPT model, KinModGPT translates the natural language descriptions of biochemical reactions into Antimony language [28], a human-readable model definition language. Next, KinModGPT converts the resulting Antimony model into the SBML model, using Tellurium [4,5]. For KinModGPT, we provided the following prompt to the GPT model. First, we told the GPT model that the task was to translate the descriptions of biochemical reactions written in natural language, into Antimony. Second, we provided a conversion rule table that showed how each chemical reaction is represented in natural language and Antimony (Figure A2). Then, we input the natural language model descriptions. The Python code of KinModGPT, and the created Antimony and SBML models, are available on GitHub (https://github.com/kmaeda16/KinModGPT, accessed on 12 April 2023).

We tested whether KinModGPT can create valid SBML models from text model descriptions. As a GPT model, we employed either of *text-davinci-003*, *gpt-3.5-turbo*, or *gpt-4*. Except for in combination with *gpt-3.5-turbo*, KinModGPT successfully generated SBML models for all the test problems (Table 1). We confirmed the validity of the generated SBML models using the Online SBML Validator. These valid SBML models could be imported by a widely used modeling tool, COPASI [6,7,8]. Moreover, we confirmed that these SBML models were consistent with their model descriptions.

To further examine the models, we intensively analyzed each generated SBML model. The decay model has a single reaction, by which the protein P degrades (Figure 1). The Antimony model is shown in Figure 1. It is worth noting that KinModGPT does not require an exact match between the conversion rules and model descriptions. Indeed, KinModGPT successfully interpreted “Protein P decays. The initial concentration is 1 uM”, as the combination of the two conversion rules shown in Figure A2: “X degrades (or decays)” and “X (concentration) is Y M (or mM or uM or nM or pM)”. The generated SBML model could be simulated as it was (Figure 3). The P concentration decreases over time.

In the model description of the HIV model (Figure 1), five sentences describe ten reactions. Indeed, the second sentence contains four reactions. KinModGPT extracted the necessary information from these complex sentences and generated a valid SBML model. The Antimony model is provided in Figure 2. Modeling tools could simulate the SBML model without any modifications. However, to check whether the SBML model reproduces the behavior of the HIV proteinase system, we manually set known realistic values to kinetic parameters. As shown in Figure 4, the substrate (S) is converted into the product (P) over time. This reaction is catalyzed by the enzyme (E). The two monomers (M) bind to form an active enzyme (E). The enzyme–substrate complex (ES) is converted into the enzyme (E) and the product (P). The enzyme (E) can bind to the product (P) and form the enzyme–product complex (EP). The enzyme (E) also binds to the inhibitor (I) and forms the enzyme–inhibitor complex (EI). This complex is then converted into the irreversible enzyme–inhibitor complex (EJ), which gradually increases. This behavior is consistent with the network map (Figure 1), model description, and literature [21,22,23].

The three-step problem highlights the remarkable capability of KinModGPT. The model description’s first two complex sentences, “Substrate S is converted into product P through intermediates M1 and M2. The metabolic reactions are catalyzed by three enzymes, E1, E2, and E3”, contain four metabolites and three enzyme reactions. These sentences may be challenging even for experienced modelers to interpret. However, combined with *text-davinci-003* or *gpt-4*, KinModGPT successfully interpreted and translated the sentences into the Antimony model (Figure 3). Next, we tested whether the created SBML model could reproduce a reasonable behavior. As shown in Figure 5, the substrate (S) is converted into the product (P) through two intermediate metabolites (M1 and M2). The expression of the third enzyme (E3), is repressed compared to the first and second enzymes (E1 and E2), because the accumulated P represses the expression of E3. This behavior matches the network map and model description shown in Figure 1.

Combined with *text-davinci-003* or *gpt-4*, KinModGPT has successfully created an SBML model for the heat shock response model, comprising 25 variables and 50 reactions, demonstrating its potential for developing complex, realistic kinetic models. The Antimony model is provided in Figure A4 and Figure A5. To test the created SBML model, we assigned realistic parameter values and simulated its behavior. Upon heat shock, proteins are rapidly denatured, and thus the yield (the fraction of folded proteins in the total protein pool) decreases (Figure 6). However, mRNA for the $\sigma^{32}$ transcription factor is activated by heat, and the produced $\sigma^{32}$ initiates the expression of the chaperone protein DnaK. Denatured proteins are then quickly refolded by DnaK, and thus the yield is recovered. This behavior is consistent with the literature [24,25,26].

### 2.3. Comparison with an Existing Tool

Gyori et al. proposed the Integrated Network and Dynamical Reasoning Assembler (INDRA), which automatically builds kinetic models from natural language texts [12]. Instead of GPT, INDRA employs DRUM (Deep Reader for Understanding Mechanisms) to interpret model descriptions. DRUM is a version of the general-purpose TRIPS natural language processing system, customized for extracting biological mechanisms from natural language text [12]. As an additional experiment, we tested whether INDRA can generate SBML models for the four test problems (Figure 1). INDRA generated valid SBML files for the decay and HIV (Table 1). However, these files were inconsistent with the model descriptions: the initial concentration was incorrect for the decay, and the SBML model for the HIV contained only one rate equation. Furthermore, INDRA failed to create SBML models for the three-step and heat shock response.

## 3. Discussion

In this article, we explored the possibilities of using GPT models for kinetic modeling automation. We developed KinModGPT by integrating GPT [19,20] and Tellurium [4,5]. KinModGPT successfully converted a kinetic model written in a natural language text into the SBML model. Furthermore, the created SBML models could be imported by widely used modeling tools. To our knowledge, this work presents the first method applying GPT models to kinetic modeling, representing an advance in systems biology.

How did GPT models fail to generate valid SBML files without any help from Tellurium? Despite the effectiveness of LLMs in generating natural sentences, their output may lack the precision required to generate accurate models. In contrast, even a single missing tag in SBML can lead to errors in the model. As a result, GPT models could not generate valid SBML models, not even for the simplest model with one variable and one reaction. Furthermore, fixing the generated invalid SBML models is challenging, because a manual review of the SBML files in XML format is required.

Instead of creating SBML models directly from natural language texts, KinModGPT employs Antimony language as an intermediate representation. Since Antimony language is simpler than SBML, GPT models can translate a natural language model description into Antimony language without errors. Then, the modeling tool, Tellurium, creates an SBML model from the Antimony model. When KinModGPT was combined with *text-davinci-003* or *gpt-4*, there were no conversion errors. However, even if there were some errors, they could be easily corrected using modeling tools or by directly editing the Antimony models. This is another advantage of KinModGPT over the GPT-only approach. Interestingly, when combined with *gpt-3.5-turbo*, KinModGPT failed to produce correct SBML models for the three-step and heat shock response. This result is reasonable, because *gpt-3.5-turbo* is tuned for chat rather than precise text generation.

Despite its promising results, KinModGPT has some limitations. Firstly, the current version of the natural language–Antimony conversion rule table, covers only a part of biochemical reactions. However, the table can easily be expanded or tailored to a specific application domain. Moreover, we may eliminate the need for manual rule definition, by fine-tuning or retraining a GPT model with a large number of “Rosetta stones”, i.e., natural language model descriptions and their Antimony counterparts. Another limitation, is that the current version of KinModGPT cannot automatically set appropriate kinetic parameter values. Thus, kinetic parameters must be tuned in the downstream modeling process, to create a realistic model that complies with experimental data [29,30,31,32,33,34].

There are many software tools available for modeling and analyzing kinetic models. However, most of them require users to manually input rate equations, which can be a significant barrier to entry. To address this issue, we have been developing tools for automating the process of kinetic modeling. For instance, we developed the CADLIVE system [9,10,11], automatically converting a biochemical network map into a kinetic model. However, even with CADLIVE, users must manually create the network map. KinModGPT is unique in its ability to generate kinetic models from natural language descriptions, and eliminates the need to draw a network map. There is another similar system, called INDRA, which automatically builds kinetic models from natural language texts [12]. While INDRA focuses on modeling cell signaling pathways, its applicability to other biochemical systems remained unexplored. Therefore, we tested if INDRA can handle metabolic pathways, protein–protein interactions, and gene regulations. As shown in the Section 2, INDRA failed to generate correct SBML models for all of the test problems (Table 1), demonstrating the superiority of KinModGPT over INDRA. Furthermore, KinModGPT offers the advantage of extensibility. Due to its simple implementation, modelers can easily customize the conversion rules for translating natural language texts to Antimony. For example, we modeled enzyme reactions using the simplest irreversible Michaelis–Menten equation in the three-step model; however, by modifying the conversion rule table (written in a simple text file), modelers can easily switch to a reversible equation. In addition, as GPT is multilingual, the conversion rules can be written in languages other than English.

KinModGPT is one of the earliest applications of an LLM to biology. The application of LLMs to biology is still in its early stages; however, LLMs will significantly impact automation in biological fields. Indeed, when augmented with external tools such as a literature search tool and a chemical synthesis planner, GPT-4, the latest version of GPT, successfully proposed a compound with similar properties to the drug dasatinib, and found a supplier that sells it [20]. We used an LLM as a natural language interpreter in the present study. This approach is applicable to different tasks in biology. For example, LLMs may be able to design DNA sequences in natural language and export them as Synthetic Biology Open Language (SBOL) [35] or GenBank files. Based on instructions such as “design an operon with a Tac promoter upstream of the gfp gene and place a BbsI recognition site before and after it”, corresponding DNA sequences can be created by LLMs. Moreover, GPT can draw pictures using scalable vector graphics (SVG) and TikZ codes [36]. In the future, LLMs may be able to draw biochemical network maps using Systems Biology Graphical Notation (SBGN) [37] or CADLIVE notation [9,11]. GPT-4 can accept both image and text inputs and produce text outputs [20]. As a result, GPT-4 has the potential to assist in the interpretation of complex bioimages. For instance, seqFISH+ can image mRNAs for 10,000 genes in single cells [38]. However, the resulting images are often too complex and information-rich for humans to comprehend. Combined with literature and gene regulatory network databases, GPT-4 and future LLMs may interpret such images and provide concise descriptions, to assist scientists. In this regard, LLMs have the potential to contribute to single-cell biology [39] and spatial transcriptomics [40], among other fields.

It should be noted, that the purpose of this article is not to provide a perfect solution for kinetic modeling automation, but to demonstrate the potential of GPT models in this field. With the continued refinement of KinModGPT, AI may be able to extract information from many relevant articles to generate a kinetic model automatically. Such a development could significantly accelerate model development and improve understanding of complex biological systems.

## 4. Materials and Methods

### 4.1. GPT

Trained on massive amounts of text data, LLMs are a powerful tool for natural language processing tasks, including language translation, text summarization, and question answering. LLMs can generate human-like natural language text, by predicting the next word in a sequence given preceding words. We employed three LLMs in this study: *text-davinci-003*, *gpt-3.5-turbo*, and *gpt-4*. The former two are in OpenAI’s GPT-3.5 series, with 175 billion parameters [19]. *gpt-3.5-turbo* is the main model used in ChatGPT [14], an improvement on *text-davinci-003*, optimized for chat. *gpt-4* [20] is the latest and most capable GPT-4 model. For GPT models, the parameter called *temperature* determines the randomness of responses. In this study, we set *temperature* = 0, for reproducibility.

### 4.2. Tellurium

Tellurium [4,5] is a Python library for modeling and simulating biochemical systems. For model development, Tellurium employs Antimony [28], a human-readable, text-based language, that facilitates the creation of kinetic models. Additionally, Tellurium can convert Antimony models to SBML models.

## Figures and Tables

**Figure 1 ijms-24-07296-f001:**
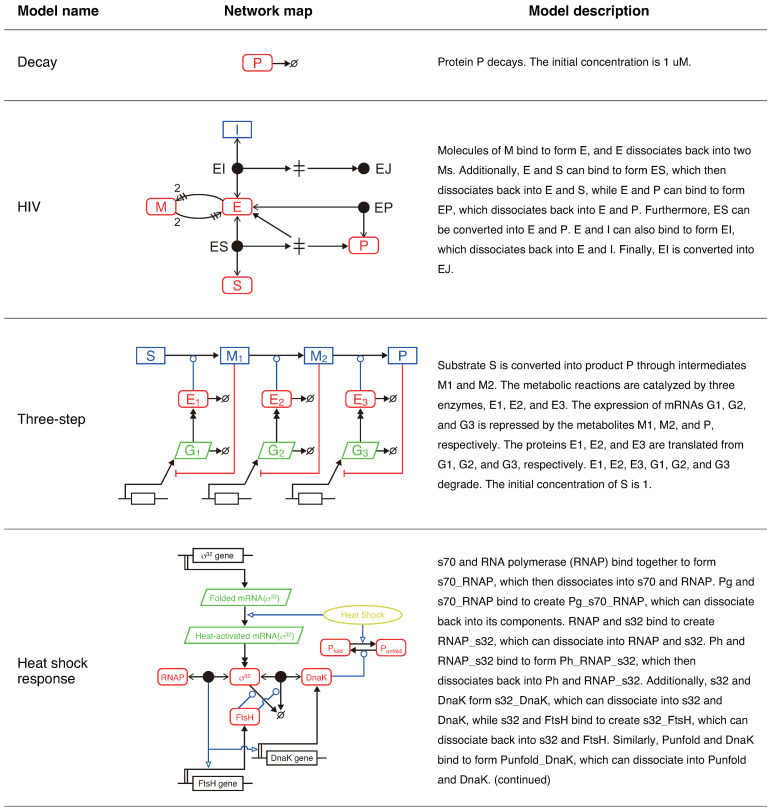
Test problems. We tested whether KinModGPT can create SBML models from the natural language model descriptions. For the reaction network maps, CADLIVE notation was used [9,10,11]. For simplicity, only important reactions are shown in the reaction network map for the heat shock response model. The complete model description for the heat shock response is provided in Figure A3.

**Figure 2 ijms-24-07296-f002:**
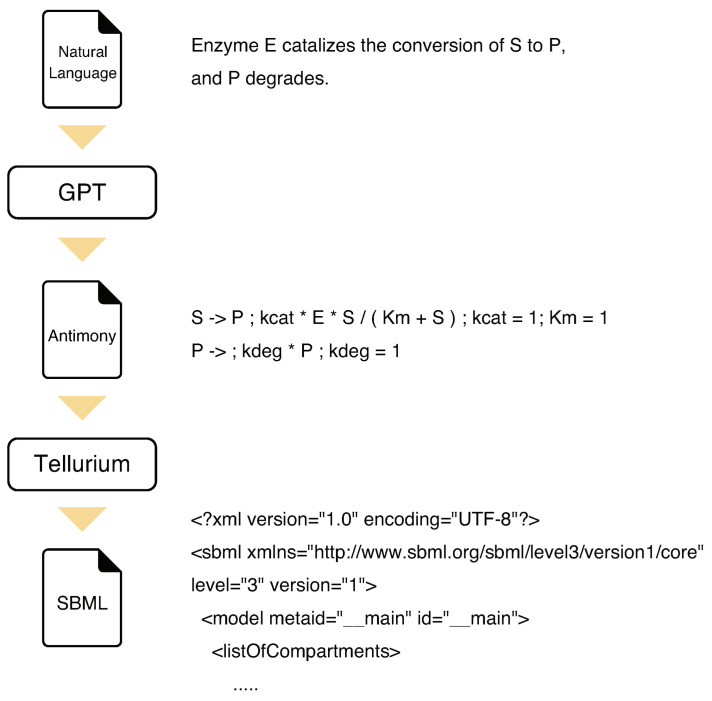
Overview of KinModGPT.

**Scheme 1 ijms-24-07296-sch001:**
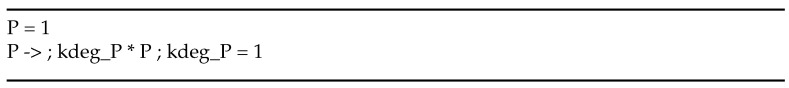
The decay model in Antimony language, created by KinModGPT with *text-davinci-003*.

**Figure 3 ijms-24-07296-f003:**
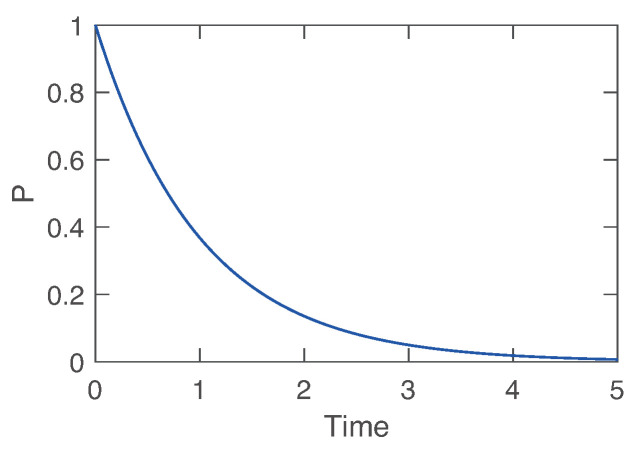
Simulation of the SBML model for the decay model. This model was created by KinModGPT with *text-davinci-003*.

**Scheme 2 ijms-24-07296-sch002:**
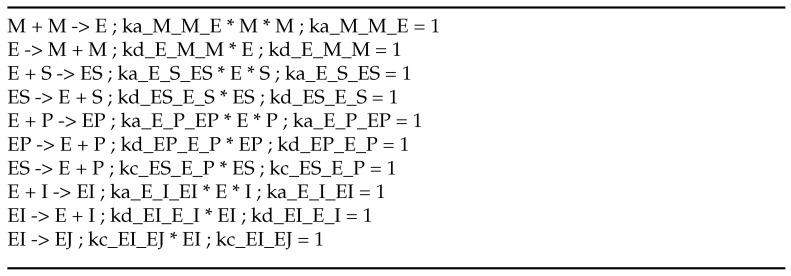
The HIV model in Antimony language, created by KinModGPT with *text-davinci-003*.

**Figure 4 ijms-24-07296-f004:**
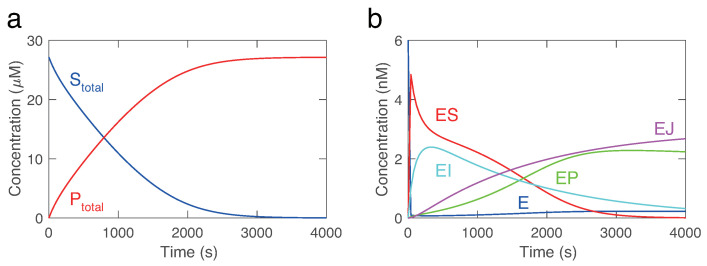
Simulation of the created SBML model for the HIV model. (**a**) $S_{total}$ and $P_{total}$ represent the total S concentration ($S_{total}=S+ES$) and the total P concentration ($P_{total}=P+EP$), respectively. (**b**) *E*, ES, EP, EI, and EJ represent the enzyme, enzyme-substrate complex, enzyme-product complex, enzyme-inhibitor complex, irreversible enzyme-inhibitor complex, respectively. Not all variables are shown, for clarity. This model was created by KinModGPT with *text-davinci-003*. We tuned the kinetic parameters before the simulation.

**Scheme 3 ijms-24-07296-sch003:**
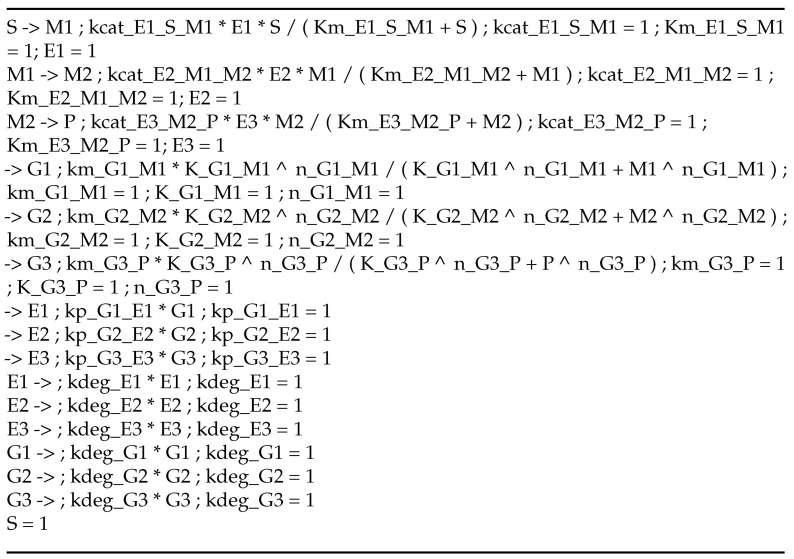
The three-step model in Antimony language, created by KinModGPT with *text-davinci-003*.

**Figure 5 ijms-24-07296-f005:**
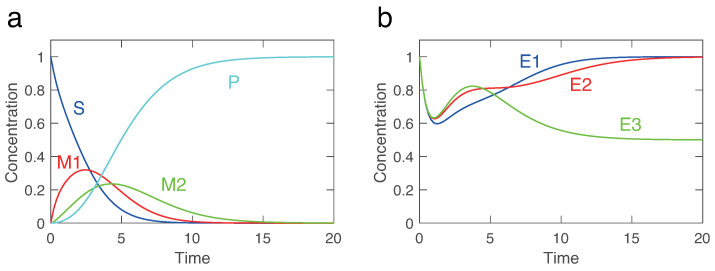
Simulation of the created SBML model for the three-step model. (**a**) S, M1, M2, and P represent the substrate, first intermediate metabolite, second intermediate metabolite, and product, respectively. (**b**) E1, E2, and E3 represent the first, second, and third enzymes, respectively. This model was created by KinModGPT with *text-davinci-003*.

**Figure 6 ijms-24-07296-f006:**
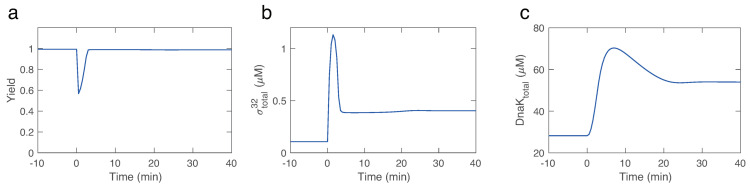
Simulation of the created SBML model for the heat shock response model. (**a**) Yield, (**b**) total $\sigma^{32}$, and (**c**) total DnaK. Heat shock occurs at 0 min and is implemented through an increase in the rate constant for protein denaturing. Yield is the fraction of folded proteins in a pool of total proteins, i.e., $Yield=P_{fold}/\left(P_{fold}+P_{unfold}+P_{unfold}\_DnaK\right)$. Not all variables are shown, for clarity. This model was created by KinModGPT with *text-davinci-003*. We tuned the kinetic parameters before the simulation.

**Table 1 ijms-24-07296-t001:** Summary of the computational experiments. We converted four natural language model descriptions into SBML, using either of the GPT-only approaches, KinModGPT approaches, or INDRA (a competitor tool). In the KinModGPT approaches, either of the three GPT models (*text-davinci-003*, *gpt-3.5-turbo*, or *gpt-4*) was employed as a natural language interpreter. The validity of the generated SBML files was verified using the Online SBML Validator [27], and their consistency with the original model descriptions was manually inspected.

Method	Model Name	Are SBML Models Created?	Are the Created SBML Models Valid?	Are the Created SBML Models Consistent with Their Model Descriptions?
*text-davinci-003* only	Decay	Yes	No	N/A
	HIV	Yes	No	N/A
	Three-step	Yes	No	N/A
	Heat shock response	Yes	No	N/A
*gpt-3.5-turbo* only	Decay	Yes	No	N/A
	HIV	Yes	No	N/A
	Three-step	Yes	No	N/A
	Heat shock response	Yes	No	N/A
*gpt-4* only	Decay	Yes	No	N/A
	HIV	Yes	No	N/A
	Three-step	Yes	No	N/A
	Heat shock response	Yes	No	N/A
KinModGPT (*text-davinci-003*)	Decay	Yes	Yes	Yes
	HIV	Yes	Yes	Yes
	Three-step	Yes	Yes	Yes
	Heat shock response	Yes	Yes	Yes
KinModGPT (*gpt-3.5-turbo*)	Decay	Yes	Yes	Yes
	HIV	Yes	Yes	Yes
	Three-step	Yes	Yes	No
	Heat shock response	No	N/A	N/A
KinModGPT (*gpt-4*)	Decay	Yes	Yes	Yes
	HIV	Yes	Yes	Yes
	Three-step	Yes	Yes	Yes
	Heat shock response	Yes	Yes	Yes
INDRA	Decay	Yes	Yes	No
	HIV	Yes	Yes	No
	Three-step	No	N/A	N/A
	Heat shock response	No	N/A	N/A

## Data Availability

The program code for the computational experiments and the created Antimony and SBML models are available on GitHub (https://github.com/kmaeda16/KinModGPT, accessed on 12 April 2023).

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
