# Peer review of "Automatic Generation of SBML Kinetic Models from Natural Language Texts Using GPT"

_ijms, 2023, doi:10.3390/ijms24087296_

Round 1
Reviewer 1 Report
The paper ins very interesting for the scientific comuunity, specially due to the application of new systems like GPT to biologycal research. However there are some facts that should be clarified in order for the text to be better understood.
In Figure 4, there are depicted S, P, and EJ which are described in the text, however there are also depicted ES, EI, EP and E but in the explanation there aren't any reference to them. I think taht a small explanation shoud be interesting.
Reviewer 2 Report
In this work, the authors proposed a novel approach called KinModGPT, which generates kinetic models directly from natural language text. KinModGPT employs GPT-3 as a natural language interpreter and Tellurium as an SBML generator. They demonstrate the effectiveness of KinModGPT in creating SBML models from complex natural language descriptions of biochemical reactions. KinModGPT successfully generates valid SBML models from a range of natural language model descriptions of metabolic pathways, protein-protein interaction networks, and heat shock response.
1) The overall writing has some formatting issues, like wording and spacing. I suggest the authors check the grammar and avoid any typos. More importantly, the writing needs improvement for readers to understand more easily.
2) As mentioned by the authors that kinetic modeling is an essential tool, which enables the quantitative analysis of biological systems. I wonder if there are such work similar with KinModGPT in this field. How KinModGPT performs compared with those similar work?
3) The modeling part are lack of details. More detailed descriptions are needed to explain the modeling part. It is not friendly for general readers.
4) The results are not quite sufficient. More discussions on the result part are needed. Moreover, I would suggest the authors discuss using the state-of-art single-cell technology (e.g., PMID: 35910046) and spatial transcriptomics (PMID: 36545790) as future perspectives, which helps expand the scope of the study.
Round 2
Reviewer 2 Report
The revision has addressed my concerns. Current version is acceptable for publication.